# Exploring the Potential of GANs in Biological Sequence Analysis

**DOI:** 10.3390/biology12060854

**Published:** 2023-06-14

**Authors:** Taslim Murad, Sarwan Ali, Murray Patterson

**Affiliations:** Department of Computer Science, Georgia State University, Atlanta, GA 30302, USA; tmurad2@student.gsu.edu (T.M.); sali85@student.gsu.edu (S.A.)

**Keywords:** sequence classification, GANs, bio-sequence analysis, class imbalance

## Abstract

**Simple Summary:**

This work deals with class imbalance issues associated with the bio-sequence datasets by employing a generative adversarial model (GAN) to improve their machine-learning-based classification performance. GAN is used to generate synthetic sequence data, which is very similar to real data in terms of tackling the data imbalance challenge. The experimental results on four distinct datasets demonstrate that GANs can improve the overall classification performance. This kind of analytical (classification) information can improve our understanding of the viruses associated with the sequences, which can be used to build prevention mechanisms to eradicate the impact of the viruses.

**Abstract:**

Biological sequence analysis is an essential step toward building a deeper understanding of the underlying functions, structures, and behaviors of the sequences. It can help in identifying the characteristics of the associated organisms, such as viruses, etc., and building prevention mechanisms to eradicate their spread and impact, as viruses are known to cause epidemics that can become global pandemics. New tools for biological sequence analysis are provided by machine learning (ML) technologies to effectively analyze the functions and structures of the sequences. However, these ML-based methods undergo challenges with data imbalance, generally associated with biological sequence datasets, which hinders their performance. Although various strategies are present to address this issue, such as the SMOTE algorithm, which creates synthetic data, however, they focus on local information rather than the overall class distribution. In this work, we explore a novel approach to handle the data imbalance issue based on generative adversarial networks (GANs), which use the overall data distribution. GANs are utilized to generate synthetic data that closely resembles real data, thus, these generated data can be employed to enhance the ML models’ performance by eradicating the class imbalance problem for biological sequence analysis. We perform four distinct classification tasks by using four different sequence datasets (Influenza A Virus, PALMdb, VDjDB, Host) and our results illustrate that GANs can improve the overall classification performance.

## 1. Introduction

Biological sequences usually refer to nucleotides or amino-acid-based sequences, and their analysis can provide detailed information about the functional and structural behaviors of the corresponding viruses, which are usually responsible for causing diseases, for example, flu [1] and COVID-19 [2]. This information is very useful in building prevention mechanisms, such as drugs [3], vaccines [4], etc., and to control the disease spread, eliminate the negative impacts, and perform virus spread surveillance.

Influenza A virus (IAV) is such an example, which is responsible for causing a highly contagious respiratory illness that can significantly threaten global public health. As the Centers for Disease Control and Prevention Center (CDC) (https://www.cdc.gov/flu/weekly/index.htm, accessed on 20 April 2023) reports, so far this season, there have been at least 25 million illnesses, 280,000 hospitalizations, and 17,000 deaths from flu in the United States. Therefore, identifying and tracking the evolution of IAV accurately is a vital step in the fight against this virus. The classification of IAV is an essential task in this aspect as it can provide valuable information on the origin, evolution, and spread of the virus. Similarly, coronaviruses are also known to infect multiple hosts and create global health crises [5] by causing pandemics, for instance, COVID-19, which is caused by the SARS-CoV-2 coronavirus. Therefore, determining the infected host information of this virus is essential for understanding the genetic diversity and evolution of the virus. As the spike protein region from the coronavirus genome is used to attach to the host cell membrane, so utilizing only the spike region provide sufficient information to determine the corresponding host. Moreover, the identification of the viral taxonomy can further enrich its understanding, e.g., the viral polymerase palmprint sequence of a virus is utilized to determine its taxonomy (species generally) [6]. A polymerase palmprint is a unique sequence of amino acids located at the thumb subunit of the viral RNA-dependent polymerase. Furthermore, examining the antigen specificities based on the T-cell receptor sequences can provide beneficial information regarding solving numerous problems of both basic and applied immunology research.

Many traditional sequence analysis methods follow phylogeny-based techniques [7,8] to identify sequence homology and predict disease transmission. However, the availability of large-size sequence data exceeds the computational limit of such techniques. Moreover, the application of ML approaches for performing biological sequence analysis is a popular research topic these days  [9,10]. The ability of ML methods to determine the sequence’s biological functions makes them desirable to be employed for sequence analysis. Additionally, ML models can also determine the relationship between the primary structure of the sequence and its biological functions. For example, ref. [9] built a random forest-based algorithm to classify the sucrose transporter (SUT) protein, ref. [10] designed a novel tool for protein–protein interactions data and functional analysis, and ref. [11] developed a new ML model to identify RNA pseudo-uridine modification sites. ML-based biological sequence analysis approaches can be categorized into feature-engineering-based methods [12,13], kernel-based methods [14], neural network-based techniques [15,16], and pre-trained deep learning models [17,18]. However, extrinsic factors limit the performance of ML-based techniques and one such major factor is data imbalance, as in the case of biological sequences, the data are generally imbalanced because the number of negative samples is much larger than that of positive samples [19]. ML models can obtain the best results when the dataset is balanced while unbalanced data will greatly affect the training of machine learning models and their application in real-world scenarios [20].

In this paper, we explore the idea of improving the performance of ML methods for biological sequence analysis by eradicating the data imbalance challenge using generative adversarial networks (GANs). Our method leverages the strengths of GANs to effectively analyze these sequences, with the potential to have significant implications for virus surveillance and tracking, as well as the development of new antiviral strategies. By accurately classifying the viral sequences, our study contributes to the field of virus surveillance and tracking. The ability to effectively identify and track viral strains can assist in monitoring the spread of infectious diseases, understanding the evolution of viruses, and informing public health interventions. Moreover, the accurate classification of the viral sequences has significant implications for the development of antiviral strategies. By better understanding the genetic diversity and relatedness of viral strains, researchers can identify potential targets for antiviral therapies, design effective vaccines, and predict the emergence of drug-resistant strains. We discuss how our study’s findings can contribute to these areas, emphasizing the importance of accurate sequence analysis in guiding the development of new antiviral strategies.

Our contributions to this work are as follows:We explore the idea of classifying biological sequences using generative adversarial networks (GANs).We show that usage of GANs improves predictive performance by eliminating the data imbalance challenge.We demonstrated the potential implications of the proposed approach for virus surveillance and tracking, and for the development of new antiviral strategies.

The rest of the paper is organized as follows: Section 2 contains the related work. The proposed approach details are discussed in Section 3. The datasets used in the experiments along with the ML models and evaluation metrics information is provided in Section 4. Section 5 highlights the experimental results and their discussion. Finally, the paper is concluded in Section 6.

## 2. Related Work

The combination of biological sequence analysis and ML models has gained quite a lot of attention among researchers in recent years [9,10]. As a biological sequence consists of a long string of characters corresponding to either nucleotides or amino acids, it needs to be transformed into a numerical form to make it compatible with the ML model. Various numerical embedding generation mechanisms are proposed to extract features from the biological sequences [12,15,18].

Some of the popular embedding generation techniques use the underlying concept of *k*-mer to compute the embeddings. Similar to how refs. [21] use the *k*-mers frequencies to obtain the vectors, refs. [13,22] combine position distribution information and *k*-mers frequencies to obtain the embeddings. Other approaches [15,16] employ neural networks to obtain the feature vectors. Moreover, kernel-based methods [14] and pre-trained deep-learning-model-based methods [17,18] also play a vital role in generating the embeddings. Although all these techniques illustrate promising analysis results, they have not mentioned anything about dealing with data imbalance issues, which if handled properly, will yield performance improvement.

Furthermore, another set of methods tackles the class imbalance challenge with the aim to enhance overall analytical performance. They use resampling techniques at the data level by either oversampling the minority class or undersampling the majority class. For instance, ref. [9] uses the borderline-SMOTE algorithm [23], an oversampling approach, to balance the feature set of the sucrose transporter (SUT) protein dataset. However, due to the usage of the k-nearest neighbor algorithm, borderline-SMOTE has high time complexity and is susceptible to noise data and is unable to make good use of the information of the majority samples [24]. Similarly, ref. [25] performs protein classification by handling the data imbalance using a hybrid sampling algorithm that combines both ensemble classifier and over-sampling techniques, KernelADASYN [26] employs a kernel-based adaptive synthetic over-sampling approach to deal with data imbalance. However, these methods do not utilize the overall data distribution, they are only based on local information [27].

## 3. Proposed Approach

In this section, we discuss our idea of exploring GANs to obtain analytical performance improvement for biological sequences in detail. As our input sequence data consists of string sequences representing amino acids, they need to be transformed into numerical representations in order to operate GANs on them. For that purpose, we use four distinct and effective numerical feature generation methods, which are described below.

### 3.1. Spike2Vec [21]

Spike2Vec generates the feature embedding by computing the *k*-mers of a sequence. As *k*-mers are known to preserve the ordering information of the sequence. *K*-mers represent a set of consecutive substrings of length *k* driven from a sequence. For *s* sequence with length *N*, the total number of its *k*-mers will be N−k+1. This method devises the feature vector for a sequence by capturing the frequencies of its *k*-mers. To further deal with the curse of dimensionality issue, Spike2Vec uses random Fourier features (RFF) to map data to a randomized low-dimensional feature space. We use k=3 to obtain the embeddings.

### 3.2. PWM2Vec [22]

This method works by using the concept of *k*-mers to obtain the numerical form of the biological sequences, however, rather than utilizing constant frequency values of the *k*-mers, it assigns weights to each amino acid of the *k*-mers and employs these weights to generate the embeddings. The position weight matrix (PWM) is used to determine the weights. PWM2Vec considers the relative importance of amino acids along with preserving the ordering information. The workflow of this method is illustrated in Figure 1 which uses k=5, while our experiments use k=3 to obtain the embeddings for performing the classification tasks.

### 3.3. Minimizer

This approach is based on the utility of minimizers [28] (*m*-mer) to obtain the feature vectors of sequences. The minimizer is extracted from a *k*-mer and it is a *m* length lexicographically smallest (in both forward and backward order) substring of consecutive alphabets from the *k*-mer. Note that m<k. The workflow of computing minimizers for a given input sequence is shown in Figure 2. This approach intends to eliminate the redundancy issue associated with *k*-mers, hence improving the storage and computation cost. Our experiments used k=9 and m=3 to generate the embeddings.

After obtaining the numerical embeddings of the biological sequences using the methods mentioned above, we further utilize these embeddings to train our GAN model. We utilize annotated groups as input to the GAN. This model has two parts, a generator model and a discriminator model. Each discriminator and generator model consists of two inner dense layers with ReLU activation functions (each followed by a batch-normalization layer) and a final dense layer. In the discriminator, the final dense layer has a Sigmoid activation function while the generator has a SoftMax activation function. The generator’s output has the same dimensions as the input data, as it synthesizes the data, while the discriminator yields a binary scalar value to indicate whether the generated data are fake or real.

The GAN model is trained using the cross-entropy loss function, ADAM optimizer, 32 batch size, and 1000 iterations. The steps followed to obtain the synthetic data after the training GAN model is illustrated in Algorithm 1. As given in the algorithm, first, the generator and discriminator models are created in steps 1–2. Then, the discriminator model is complied for training with cross-entropy loss and ADAM optimizer in step 3. After that, the count and length of synthetic sequences along with the number of training epochs and batch size are mentioned in steps 4–6. Then, the training of the models occurs in steps 7–12 , where each of the models is fine-tuned for the given number of iterations. Once the GAN model is trained, its generator part is employed to synthesize new embedding data which resemble real-world data. These synthesized data can eliminate the data imbalance problem, improving the analytical performance. Moreover, the overall workflow of training the GAN model is shown in Figure 3. The figure illustrates the training procedure of the GAN model by fine-tuning the parameters of its generator and discriminator modules. It starts by obtaining the numerical embeddings of the input sequences and passing them to the discriminator part along with the synthetic data generated by the generator part. The discriminator model is trained in a way that it can identify whether the data are real or synthetic, and based on this information, we fine-tune the generator model. The overall goal is training the generator model to the extent that the synthetic data generated by it cannot be distinguished by the discriminator model anymore, which means that the synthetic data are very close to the real data.
**Algorithm 1** Training GAN model       **Input:** Set of Sequences *S*, ganCnt       **Output:** GANs based sequences $S^{\prime }$1: m_gen←generator()▹ generator model2: m_dis←discriminator()▹ discriminator model3: m_dis.compile(loss=CE,opt=ADAM)
4: seqLen←len(S[0])▹ len of each $S^{\prime }$ sequence5: iter←1000
6: batch_size←32
7: **for** *i* in iter **do**
8:       noise←random(ganCnt,seqLen)
9:       $S^{\prime }\leftarrow m\_gen.predict(noise)$▹ get GAN sequences10:     m_dis.backward(m_dis.loss)▹ fine-tune m_dis11:     m_gen.backward(m_gen.loss)▹ fine-tune m_gen12: **end for**13: return($S^{\prime }$)

## 4. Experimental Setup

This section highlights the details of the datasets used to conduct the experiments along with the information about the classification models and their respective evaluation metrics to report the performance. All experiments were carried out on an Intel (R) Core i5 system with a 2.40 GHz processor and 32 GB memory. We use Python to run the experiments. Our code and preprocessed datasets are available online for reproducibility (https://github.com/taslimmurad-gsu/GANs-Bio-Seqs/tree/main, accessed on 20 April 2023).

### 4.1. Dataset Statistics

We use 4 different datasets to evaluate our suggested method. A detailed description of each of the dataset is given as follows.

#### 4.1.1. Influenza A Virus

We are using the influenza A virus sequence dataset belonging to two kinds of subtypes “H1N1” and “H3N2” extracted from [29] website. These data contain 222,450 sequences in total with 119,100 sequences belonging to the H1N1 subtype and 103,350 to the H2N3 subtype. The detailed statistics for this dataset are shown in Table 1. We use these two subtypes as labels to classify the Influenza A virus in our experiments.

#### 4.1.2. PALMdb

The PALMdb [6,30] dataset consists of viral polymerase palmprint sequences, which can be classified species-wise. This dataset is created by mining the public sequence databases using the palmscan [6] algorithm. It has 124,908 sequences corresponding to 18 different virus species. The distribution of these species is given in Table 2 and more detailed statistics are shown in Table 1. We use the species name as a label to do the classification of the PALMdb sequences.

#### 4.1.3. VDjDB

VDJdb is a curated dataset of T-cell receptor (TCR) sequences with known antigen specificities [31]. This dataset consists of 58,795 human TCRs and 3353 mouse TCRs. More than half of the examples are TRBs (*n* = 36,462) with the remainder being TRAs (*n* = 25,686). The T-cell receptor alpha chain (TRA) and T-cell receptor beta chain (TRB) refer to the chains that make up the T-cell receptor (TCR) complex. The TRB chain plays a crucial role in antigen recognition and is involved in T-cell immune responses. It has 78,344 total sequences belonging to 17 unique antigen species. The distribution of the sequence among the antigen species is shown in Table 3 and further details of the dataset are given in Table 1. We use these data to perform the antigen species classification.

#### 4.1.4. Coronavirus Host

The host dataset consists of spike sequences of coronavirus corresponding to various infected hosts. These data are extracted from ViPR [32] and GISAID [33]. They contain 5558 total sequences belonging to 21 unique hosts and their detailed distribution is shown in Table 4.

### 4.2. ML Classifiers and Evaluation Metrics

To perform classification tasks, we employed the following ML models: naive Bayes (NB), multilayer perceptron (MLP), *k*-nearest neighbor (*k*-NN) (where k=3), random forest (RF), logistic regression (LR), and decision tree (DT). For each classification task, the data are split into 30–70% train–test sets using stratified sampling to preserve the original data distribution. Furthermore, our experiments were conducted by averaging the performance results of 5 runs for each combination of dataset and classifier to obtain more stable results.

We evaluated the classifiers using the following performance metrics: accuracy, precision, recall, weighted F1, F1 macro, and ROC AUC macro. Since we are doing multi-class classification in some cases, we utilized the one-vs-rest approach for computing the ROC AUC score for them. Moreover, the reason for reporting many metrics is to obtain more insight into the classifiers’ performance, especially in the class imbalance scenario where reporting only accuracy does not provide sufficient performance information.

## 5. Results and Discussion

This section discusses the experimental results comprehensively. The subtype classification results of the Influenza A virus dataset are given in Table 5, along with the results of the PALMdb dataset species-wise classification. The antigen species-wise classification results of VDjDB data and host-wise classification results of coronavirus host data are shown in Table 6. The reported results represent the results achieved using the test set.

We have compared the classification performance of three embedding generation methods (Spike2Vec, PWM2Vec, Min2Vec) using four datasets (Influenza A virus, PALMdb, VDjDb, Host) under three different settings (without-GANs, with-GANs, only-GANs). Without-GANs indicate the scenario where the original embeddings from the three embedding generation methods are used to perform the classifications, while with-GANs show the performance achieved using the original embeddings with the addition of the GANs-based synthetic data for eliminating the class imbalance challenge. The only-GANs setting is utilized to illustrate the performance gained by using only the synthetic data without the original one. It provides an overview of the effectiveness of the synthetic data in terms of classification predictive performance.

In the with-GANs scenario, for each dataset, the classes with a lower number of instances combine their respective GANs-based synthetic data to increase their count to make them comparable with the most frequent classes. This addition removes the data imbalance issue and the newly created dataset is further utilized for performing the classification tasks. Note that the synthetic data are only added to the training set, while the test set contains the original data, so the test set has the actual imbalance data distribution. A further detailed discussion of the results for each embedding method with various combinations of datasets and setting scenarios are given below.

### 5.1. Performance of without-GANs Data

These results illustrate the classification performance achieved corresponding to the embeddings generated by Spike2Vec, PWM2Vec, and minimizer strategies for each dataset. We can observe that for the Influenza A virus dataset, Spike2Vec and minimizer are exhibiting similar performance for almost all the classifiers and are better than PWM2Vec. However, the NB model yields minimum predictive performance for all the embeddings. Similarly, the VDjDb dataset portrays similar performance for Spike2Vec and minimizer for all evaluation metrics, while its PWM2Vec has a very low predictive performance. Moreover, all the embeddings achieve the same performance in terms of all the evaluation metrics for every classifier on the PALMdb dataset. For the host dataset, all the three embeddings are yielding very similar results with NB exhibiting the lowest and RF exhibiting the highest performances.

### 5.2. Performance of with-GANs Data

To view the impact of GAN-based data on the predictive performance for all the datasets, we evaluate the performance using the original embeddings with GAN-based synthetic data added to them, respectively. These GANs-based data are used to train the classifiers, while only the original data are used as test data. For a dataset, to generate the GAN data corresponding to an embedding generation method, the GAN model is trained with the original embeddings first and then new data are synthesized for that embedding. Every label of the embedding will have a different count of synthetic data added to it depending on its count in the original embedding data. The aim is to make the class distribution balanced in a dataset.

For Influenza A virus data, the results show that in some cases the addition of GANs-based synthetic data improves the performance as compared to the performance on the original data, such as for the KNN, RF, and NB classifiers corresponding to PWM2Vec methods. Similarly, on the VDjDB dataset, the GAN-based improvement is also witnessed in some cases, such as for all the classifiers corresponding to the PWM2Vec method except NB. Moreover, as the performance of the PALMdb dataset on the original data is at its maximum already, the addition of GAN embeddings has retained that performance. Furthermore, the host dataset combining the synthetic data with the original data shows a performance improvement for some scenarios; for instance, PWM2Vec-based classification using NB, KNN, and RF classifiers, Spike2Vec- and Min2Vec-based classifications using NB and KNN classifiers.

Generally, we can observe that the inclusion of GAN synthetic data in the training set can improve the overall classification performance. This is because the training set size increases and the data imbalance issue is resolved by adding the respective synthetic data.

### 5.3. Performance of Only-GANs Data

We also studied the classification performance gain of using only GANs-based embeddings without the original data. The results depict that for all four datasets, this category has the lowest predictive performance for all the combinations of classifiers and embeddings as compared to the performance on original data and on original data with GANs. As only the synthetic data are employed to train the classifiers, they are tested on the original data, which is why the performance is low as compared to others.

### 5.4. Data Visualization

We visualize our datasets using the popular visualization technique, *t*-SNE [34], to view the internal structure of each dataset following various embeddings. The plots for the Influenza A virus dataset are reported in Figure 4. We can observe that for Spike2Vec and minimizer-based plots, the addition of GAN-based features causes two big clusters along with the small scattered clusters for each, unlike their original *t*-SNEs, which only consist of small scattered groups. However, the PWM2Vec-based plots for both with GANs and without GANs show similar structures; however, generally including GAN-based embeddings to the original ones can improve the *t*-SNE structure.

Similarly, the *t*-SNE plots for the PALMdb dataset corresponding to different embeddings are shown in Figure 5. We can observe that this dataset shows similar kinds of cluster patterns corresponding to both without-GANs- and with-GANs-based embeddings. As the original dataset already shows clear and distinct clusters for various species, adding GAN-based embedding to it does not affect the cluster structure much.

Moreover, the *t*-SNE plots for the VDjDB dataset are given in Figure 6. We can observe that the addition of GAN-based features to the minimizer-based embedding has yielded more clear and distinct clusters in the visualization. GAN-based spike2vec also portrays more clusters than the Spike2Vec one. However, the PWM2Vec shows similar patterns for both GAN-based and without GANs embeddings. Overall, it indicates that adding GANs-based features is enhancing the *t*-SNE cluster structures.

Furthermore, the *t*-SNE plots for the host dataset are illustrated in Figure 7. We can see that for PWM2Vec, the addition of GANs-based embeddings further refines the structure by reshaping the clusters, while the structures of Spike2Vec and Min2Vec seem to remain almost same for both with and without GANs.

We also investigated the *t*-SNE structures generated by using only the GANs-based embeddings and Figure 8 illustrates the results. It can be seen that for all the datasets only-GAN embeddings are yielding non-overlapping distinct clusters corresponding to each group with respect to the dataset. It is because, for each group, the only-GAN embeddings are synthesized after training the GAN model with the original data of the respective group. Note that for host data, some of the clusters are very tiny because of the corresponding number of instances in the dataset belonging to that group being very small.

## 6. Conclusions

In conclusion, this work explores a novel approach to improve the predictive performance of the biological sequence classification task by using GANs. It generates synthetic data with the help of GANs to eliminate the data imbalance challenge, hence improving the performance. In the future, we would like to extend this study by investigating more advanced variations of GANs to synthesize the biological sequences and their impacts on the biological sequence analysis. We also want to examine additional genetic data, such as hemagglutinin and neuraminidase gene sequences, with GANs to improve their classification accuracy.

## Figures and Tables

**Figure 1 biology-12-00854-f001:**
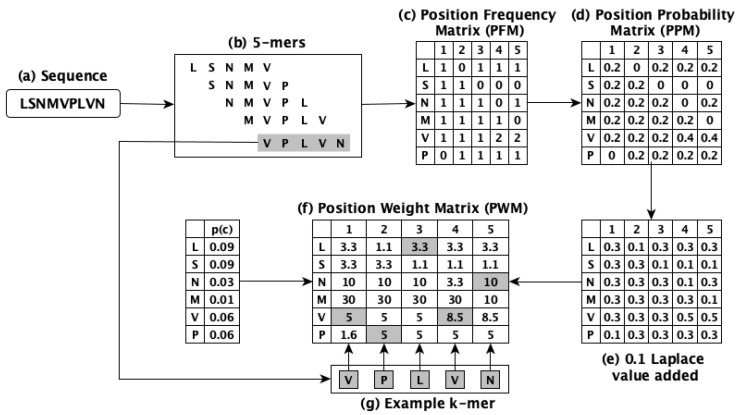
The workflow of PWM2Vec method for a given sequence.

**Figure 2 biology-12-00854-f002:**
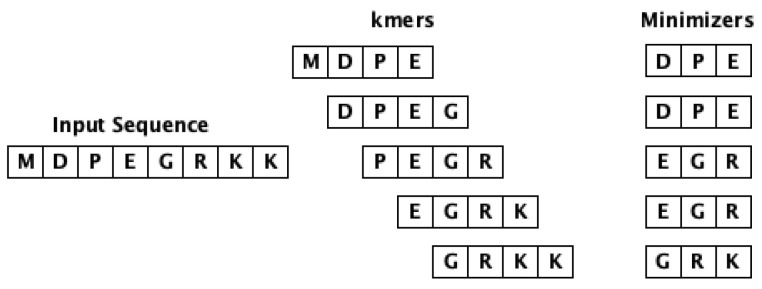
The workflow of obtaining minimizers from an input sequence.

**Figure 3 biology-12-00854-f003:**
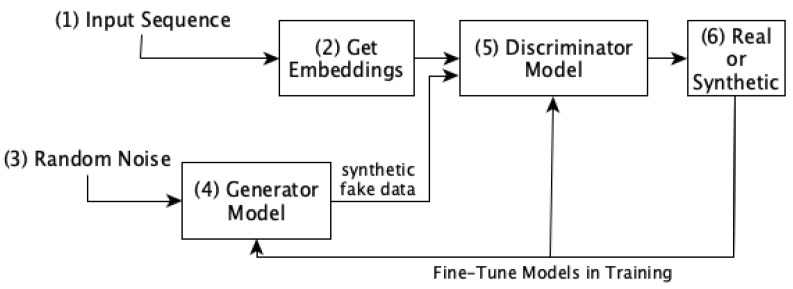
The workflow of training the GAN model. It shows the process followed to fine tune the parameters of the generator and discriminator models while training.

**Figure 4 biology-12-00854-f004:**
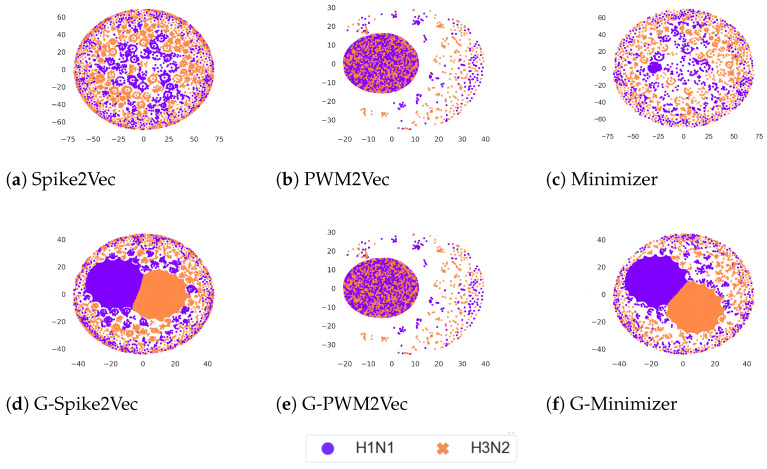
*t*-SNE plots for Influenza A virus dataset without GANs (**a**–**c**) and with GANs (**d**–**f**). The figure is best seen in color.

**Figure 5 biology-12-00854-f005:**
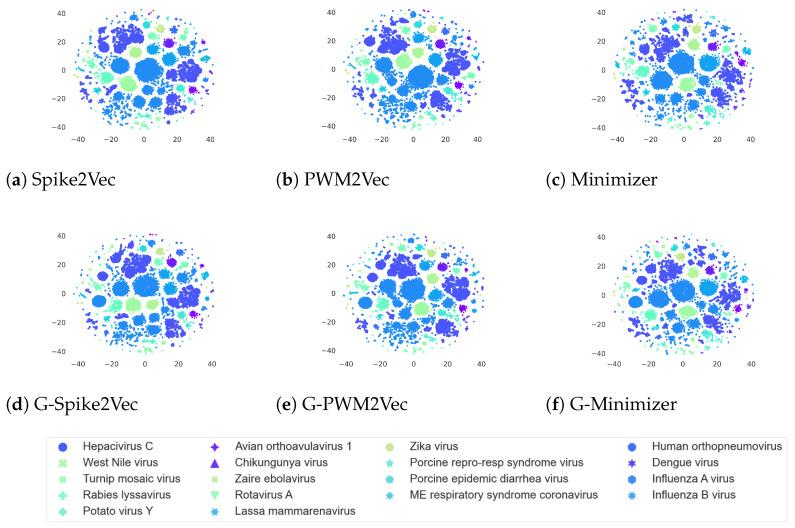
*t*-SNE plots for PALMdb dataset without GANs (**a**–**c**), and with GANs (**d**–**f**). The figure is best seen in color.

**Figure 6 biology-12-00854-f006:**
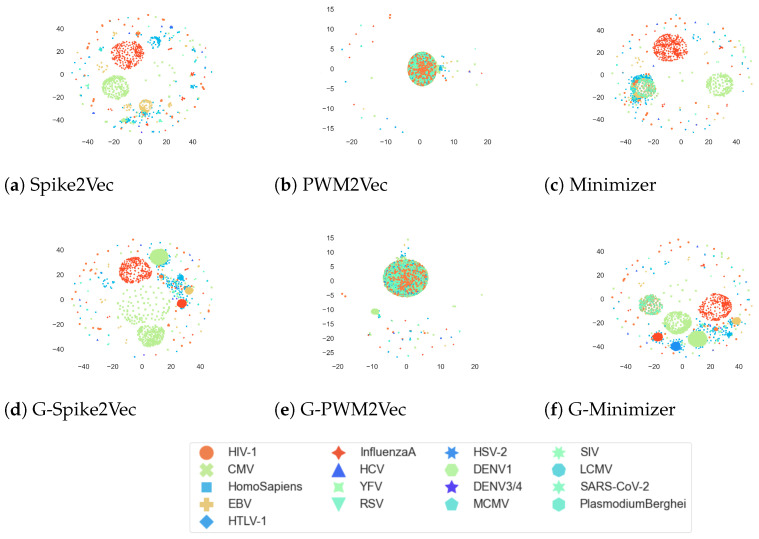
*t*-SNE plots for VDjDB dataset without GANs (**a**–**c**), and with GANs (**d**–**f**). The figure is best seen in color.

**Figure 7 biology-12-00854-f007:**
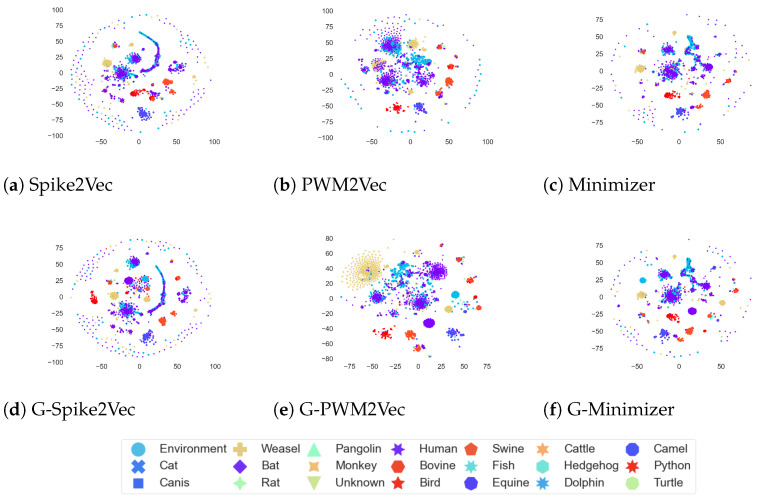
*t*-SNE plots for Host dataset without GANs (**a**–**c**) and with GANs (**d**–**f**). The figure is best seen in color.

**Figure 8 biology-12-00854-f008:**
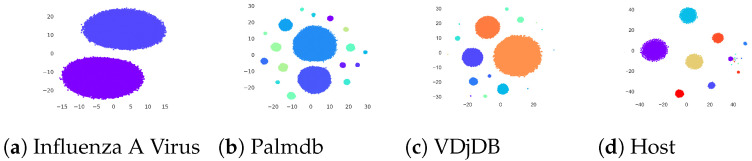
*t*-SNE plots of only GANs embeddings for Influenza A virus, PALMdb, VDjDB, and host datasets. The figure is best seen in color.

**Table 1 biology-12-00854-t001:** Dataset Statistics of each dataset used in our experiments.

				Sequence Length		
Name	|Sequences|	Classes	Goal	Min	Max	Average
Influenza A Virus	222,450	2	Virus Subtypes Classification	11	71	68.60
PALMdb	124,908	18	Virus Species Classification	53	150	130.83
VDjDB	78,344	17	Antigen Species Classification	7	20	12.66
Host	5558	21	Coronavirus Host Classification	9	1584	1272.36

**Table 2 biology-12-00854-t002:** Species-wise distribution of PALMdb dataset.

Species Name	Count	Species Name	Count
Avian orthoavulavirus 1	2353	Chikungunya virus	2319
Dengue virus	1627	Hepacivirus C	29,448
Human orthopneumovirus	3398	Influenza A virus	47,362
Influenza B virus	8171	Lassa mammarenavirus	1435
Middle East respiratory syndrome-related coronavirus	1415	Porcine epidemic diarrhea virus	1411
Porcine reproductive and respiratory syndrome virus	2777	Potato virus Y	1287
Rabies lyssavirus	4252	Rotavirus A	4214
Turnip mosaic virus	1109	West Nile virus	5452
Zaire ebolavirus	4821	Zika virus	2057

**Table 3 biology-12-00854-t003:** Antigen species-wise distribution of VDjDB dataset.

Antigen Species Name	Count	Antigen Species Name	Count
CMV	37,357	DENV1	180
DENV3/4	177	EBV	11,026
HCV	840	HIV-1	3231
HSV-2	154	HTLV-1	232
HomoSapiens	4646	InfluenzaA	14,863
LCMV	141	MCMV	1463
PlasmodiumBerghei	243	RSV	125
SARS-CoV-2	758	SIV	2119
YFV	789		

**Table 4 biology-12-00854-t004:** Coronavirus host-wise distribution of host dataset.

Host Name	Count	Host Name	Count
Bat	153	Bird	374
Bovine	88	Camel	297
Canis	40	Cat	123
Cattle	1	Dolphin	7
Environment	1034	Equine	5
Fish	2	Hedgehog	15
Human	1813	Monkey	2
Pangolin	21	Python	2
Rat	26	Swine	558
Turtle	1	Unknown	2
Weasel	994	-	-

**Table 5 biology-12-00854-t005:** The subtype classification results of Influenza A virus dataset and species-wise classification results of Palmdb dataset. These results are average results over five runs. The best value for each metric is shown as bold.

	Method		Influenza A Virus								PALMdb						
		Algo.	Acc. ↑	Prec. ↑	Recall ↑	F1 (Weig.) ↑	F1 (Macro) ↑	ROC AUC ↑	Train Time (Sec.) ↓		Acc. ↑	Prec. ↑	Recall ↑	F1 (Weig.) ↑	F1 (Macro) ↑	ROC AUC ↑	Train Time (Sec.) ↓
Without GANs	Spike2Vec [21]	NB	0.538	0.673	0.538	0.382	0.358	0.503	96.851		**0.999**	**0.999**	**0.999**	**0.999**	**0.999**	**0.999**	453.961
		MLP	**0.999**	**0.999**	**0.999**	**0.999**	**0.999**	**0.999**	742.551		**0.999**	**0.999**	**0.999**	**0.999**	**0.999**	**0.999**	1446.421
		KNN	**0.999**	**0.999**	**0.999**	**0.999**	**0.999**	**0.999**	2689.320		**0.999**	**0.999**	**0.999**	**0.999**	**0.999**	**0.999**	1274.75
		RF	**0.999**	**0.999**	**0.999**	**0.999**	**0.999**	**0.999**	433.459		**0.999**	**0.999**	**0.999**	**0.999**	**0.999**	**0.999**	166.087
		LR	0.966	0.966	0.966	0.966	0.966	0.965	24.467		**0.999**	**0.999**	**0.999**	**0.999**	**0.999**	**0.999**	31,564.898
		DT	**0.999**	**0.999**	**0.999**	**0.999**	**0.999**	**0.999**	54.024		**0.999**	**0.999**	**0.999**	**0.999**	**0.999**	**0.999**	163.827
	PWM2Vec [22]	NB	0.563	0.745	0.563	0.435	0.414	0.530	60.155		**0.999**	**0.999**	**0.999**	**0.999**	**0.999**	**0.999**	562.922
		MLP	0.644	0.785	0.644	0.579	0.566	0.617	1471.086		**0.999**	**0.999**	**0.999**	**0.999**	**0.999**	**0.999**	1675.896
		KNN	0.644	0.785	0.644	0.579	0.566	0.617	2665.538		**0.999**	**0.999**	**0.999**	0.998	**0.999**	**0.999**	1514.240
		RF	0.644	0.785	0.644	0.579	0.566	0.618	1514.979		**0.999**	**0.999**	**0.999**	**0.999**	**0.999**	**0.999**	284.450
		LR	0.644	0.784	0.644	0.579	0.565	0.617	388.235		**0.999**	**0.999**	**0.999**	**0.999**	**0.999**	**0.999**	41,029.833
		DT	0.644	0.785	0.644	0.579	0.566	0.617	78.525		**0.999**	**0.999**	**0.999**	**0.999**	**0.999**	**0.999**	233.533
	Minimizer	NB	0.679	0.682	0.679	0.673	0.669	0.670	57.469		**0.999**	**0.999**	**0.999**	**0.999**	**0.999**	**0.999**	474.482
		MLP	0.998	0.998	0.998	0.998	0.998	0.998	1864.844		**0.999**	**0.999**	**0.999**	**0.999**	**0.999**	**0.999**	3958.188
		KNN	**0.999**	**0.999**	**0.999**	**0.999**	**0.999**	**0.999**	2818.292		**0.999**	**0.999**	**0.999**	**0.999**	**0.999**	**0.999**	1357.673
		RF	**0.999**	**0.999**	**0.999**	**0.999**	**0.999**	**0.999**	1039.824		**0.999**	**0.999**	**0.999**	**0.999**	**0.999**	**0.999**	399.507
		LR	0.719	0.719	0.719	0.719	0.718	0.718	186.522		**0.999**	**0.999**	**0.999**	**0.999**	**0.999**	**0.999**	7270.111
		DT	**0.999**	**0.999**	**0.999**	**0.999**	**0.999**	**0.999**	72.510		**0.999**	**0.999**	**0.999**	**0.999**	**0.999**	**0.999**	223.215
With GANs	Spike2Vec [21]	NB	0.538	0.681	0.538	0.380	0.355	0.502	138.179		**0.999**	**0.999**	**0.999**	**0.999**	**0.999**	**0.999**	197.033
		MLP	0.992	0.992	0.992	0.992	0.992	0.992	1604.287		**0.999**	**0.999**	**0.999**	**0.999**	**0.999**	**0.999**	491.182
		KNN	**0.999**	**0.999**	**0.999**	**0.999**	**0.999**	**0.999**	3546.211		**0.999**	**0.999**	**0.999**	**0.999**	**0.999**	**0.999**	689.672
		RF	**0.999**	**0.999**	**0.999**	**0.999**	**0.999**	**0.999**	784.393		**0.999**	**0.999**	**0.999**	**0.999**	**0.999**	**0.999**	243.646
		LR	0.957	0.957	0.957	0.957	0.957	0.957	6810.398		**0.999**	**0.999**	**0.999**	**0.999**	**0.999**	**0.999**	2643.646
		DT	**0.999**	**0.999**	**0.999**	**0.999**	**0.999**	**0.999**	365.332		**0.999**	**0.999**	**0.999**	**0.999**	**0.999**	**0.999**	396.362
	PWM2Vec [22]	NB	0.565	0.748	0.565	0.437	0.416	0.532	107.617		**0.999**	**0.999**	**0.999**	**0.999**	**0.999**	**0.999**	569.510
		MLP	0.644	0.784	0.644	0.579	0.566	0.617	1817.859		**0.999**	**0.999**	**0.999**	**0.999**	**0.999**	**0.999**	1337.920
		KNN	0.646	0.785	0.646	0.581	0.568	0.619	2965.701		**0.999**	**0.999**	**0.999**	**0.999**	**0.999**	**0.999**	1524.009
		RF	0.646	0.786	0.646	0.582	0.569	0.619	1837.425		**0.999**	**0.999**	**0.999**	**0.999**	**0.999**	**0.999**	1802.577
		LR	0.632	0.793	0.632	0.589	0.597	0.657	10,273.672		**0.999**	**0.999**	**0.999**	**0.999**	**0.999**	**0.999**	3549.095
		DT	0.646	0.786	0.646	0.581	0.568	0.619	1264.188		**0.999**	**0.999**	**0.999**	**0.999**	**0.999**	**0.999**	2580.831
	Minimizer	NB	0.611	0.726	0.611	0.534	0.520	0.584	127.058		**0.999**	**0.999**	**0.999**	**0.999**	**0.999**	**0.999**	669.513
		MLP	0.976	0.976	0.976	0.976	0.976	0.976	825.868		**0.999**	**0.999**	**0.999**	**0.999**	**0.999**	**0.999**	1231.650
		KNN	**0.999**	**0.999**	**0.999**	**0.999**	**0.999**	**0.999**	3163.325		**0.999**	**0.999**	**0.999**	**0.999**	**0.999**	**0.999**	1484.555
		RF	**0.999**	**0.999**	**0.999**	**0.999**	**0.999**	**0.999**	1557.065		**0.999**	**0.999**	**0.999**	**0.999**	**0.999**	**0.999**	1699.503
		LR	0.711	0.712	0.711	0.711	0.710	0.711	2179.485		**0.999**	**0.999**	**0.999**	**0.999**	**0.999**	**0.999**	3482.345
		DT	**0.999**	**0.999**	**0.999**	**0.999**	**0.999**	**0.999**	481.232		**0.999**	**0.999**	**0.999**	**0.999**	**0.999**	**0.999**	2700.860
Only GANs For Training	Spike2Vec [21]	NB	0.443	0.318	0.443	0.296	0.317	0.476	69.293		0.056	0.005	0.056	0.009	0.014	0.523	172.517
		MLP	0.499	0.506	0.499	0.498	0.499	0.503	279.364		0.104	0.260	0.104	0.148	0.039	0.486	264.306
		KNN	0.586	0.623	0.586	0.523	0.510	0.561	4088.144		0.126	0.242	0.126	0.156	0.123	0.533	263.101
		RF	0.464	0.215	0.464	0.294	0.317	0.500	386.409		0.011	0.000	0.011	0.000	0.001	0.500	8451.755
		LR	0.523	0.523	0.523	0.523	0.520	0.520	469.512		0.001	0.000	0.001	0.000	0.001	0.500	1481.505
		DT	0.535	0.286	0.535	0.373	0.348	0.500	308.698		0.042	0.001	0.042	0.003	0.004	0.499	2764.815
	PWM2Vec [22]	NB	0.468	0.508	0.468	0.331	0.351	0.500	60.008		0.034	0.004	0.034	0.003	0.004	0.499	370.330
		MLP	0.471	0.503	0.471	0.369	0.385	0.500	333.503		0.400	0.335	0.400	0.355	0.080	0.534	577.936
		KNN	0.520	0.575	0.520	0.470	0.480	0.542	4565.427		0.061	0.213	0.061	0.089	0.059	0.496	2475.871
		RF	0.535	0.286	0.535	0.372	0.348	0.500	746.999		0.034	0.001	0.034	0.002	0.003	0.500	10,880.182
		LR	0.534	0.603	0.534	0.482	0.492	0.557	975.877		0.001	0.012	0.001	0.009	0.009	0.490	278.851
		DT	0.535	0.286	0.535	0.372	0.348	0.500	500.541		0.022	0.001	0.022	0.032	0.013	0.500	3078.085
	Minimizer	NB	0.523	0.529	0.523	0.523	0.523	0.526	65.955		0.062	0.194	0.062	0.048	0.055	0.525	497.483
		MLP	0.477	0.495	0.477	0.447	0.455	0.494	499.569		0.005	0.003	0.005	0.003	0.008	0.475	707.236
		KNN	0.539	0.538	0.539	0.538	0.535	0.536	5211.216		0.177	0.155	0.177	0.148	0.058	0.522	3116.525
		RF	0.535	0.287	0.535	0.373	0.348	0.499	624.564		0.034	0.001	0.034	0.002	0.003	0.500	10,349.430
		LR	0.548	0.548	0.548	0.548	0.546	0.546	771.273		0.201	0.120	0.201	0.228	0.102	0.501	3234.386
		DT	0.464	0.215	0.464	0.294	0.317	0.500	576.693		0.003	0.002	0.003	0.002	0.003	0.500	346.660

**Table 6 biology-12-00854-t006:** The antigen species-wise classification results of VDjDB dataset and host-wise classification results of coronavirus host dataset. These results are average values over five runs. The best value for each metric is shown as bold.

	Method		VDjDB								Host						
		Algo.	Acc. ↑	Prec. ↑	Recall ↑	F1 (Weig.) ↑	F1 (Macro) ↑	ROC AUC ↑	Train Time (Sec.) ↓		Acc. ↑	Prec. ↑	Recall ↑	F1 (Weig.) ↑	F1 (Macro) ↑	ROC AUC ↑	Train Time (Sec.) ↓
Without GANs	Spike2Vec [21]	NB	**0.999**	**0.999**	**0.999**	**0.999**	**0.999**	**0.999**	87.948		0.664	0.752	0.664	0.647	0.578	0.787	9.735
		MLP	**0.999**	**0.999**	**0.999**	**0.999**	**0.999**	**0.999**	689.357		0.824	0.832	0.824	0.822	0.702	0.839	117.344
		KNN	0.998	0.998	0.998	0.998	0.998	**0.999**	167.426		0.776	0.820	0.776	0.791	0.642	0.824	2.421
		RF	**0.999**	**0.999**	**0.999**	**0.999**	**0.999**	**0.999**	152.581		**0.849**	**0.851**	**0.849**	**0.851**	0.701	0.851	22.611
		LR	**0.999**	**0.999**	**0.999**	**0.999**	**0.999**	**0.999**	882.695		0.843	0.850	0.843	0.840	0.683	**0.880**	2169.697
		DT	**0.999**	**0.999**	**0.999**	**0.999**	**0.999**	**0.999**	43.314		0.828	0.830	0.828	0.833	0.651	0.844	6.415
	PWM2Vec [22]	NB	0.179	0.926	0.179	0.250	0.305	0.634	84.292		0.436	0.625	0.436	0.396	0.433	0.709	2.782
		MLP	0.525	0.685	0.525	0.399	0.315	0.626	1216.913		0.797	0.808	0.797	0.787	0.606	0.815	69.551
		KNN	0.525	0.689	0.525	0.399	0.320	0.626	248.660		0.795	0.792	0.795	0.789	0.647	0.816	0.917
		RF	0.525	0.690	0.525	0.400	0.320	0.626	736.583		0.833	0.834	0.833	0.827	0.691	0.853	10.166
		LR	0.525	0.681	0.525	0.400	0.320	0.626	299.575		0.802	0.817	0.802	0.794	0.671	0.843	693.437
		DT	0.525	0.690	0.525	0.400	0.320	0.626	39.697		0.803	0.804	0.803	0.800	0.625	0.826	7.063
	Minimizer	NB	0.930	0.972	0.930	0.940	0.838	0.935	98.159		0.480	0.690	0.480	0.452	0.557	0.753	18.977
		MLP	0.934	0.952	0.934	0.928	0.782	0.882	1253.018		0.782	0.797	0.782	0.774	0.677	0.831	279.057
		KNN	0.951	0.961	0.951	0.947	0.849	0.925	172.851		0.763	0.786	0.763	0.765	0.688	0.832	4.831
		RF	0.953	0.962	0.953	0.948	0.847	0.927	468.139		0.835	0.840	0.835	0.827	0.705	0.843	60.184
		LR	0.952	0.961	0.952	0.948	0.847	0.926	203.061		0.818	0.827	0.818	0.811	0.693	0.839	978.112
		DT	0.952	0.962	0.952	0.948	0.847	0.926	25.392		0.818	0.824	0.818	0.813	0.683	0.841	4.959
With GANs	Spike2Vec [21]	NB	**0.999**	**0.999**	**0.999**	**0.999**	0.988	**0.999**	78.891		0.684	0.759	0.684	0.664	0.656	0.843	12.607
		MLP	**0.999**	**0.999**	**0.999**	**0.999**	**0.999**	**0.999**	1085.850		0.747	0.772	0.747	0.741	0.482	0.790	187.038
		KNN	0.998	0.998	0.998	0.998	0.992	0.998	135.567		0.796	0.797	0.796	0.791	0.638	0.826	3.193
		RF	**0.999**	**0.999**	**0.999**	**0.999**	**0.999**	**0.999**	186.662		**0.849**	**0.851**	**0.849**	0.842	0.707	0.852	30.672
		LR	**0.999**	**0.999**	**0.999**	**0.999**	**0.999**	**0.999**	5736.169		0.826	0.840	0.826	0.821	0.726	0.873	4897.412
		DT	**0.999**	**0.999**	**0.999**	**0.999**	**0.999**	**0.999**	143.618		0.814	0.819	0.814	0.811	0.678	0.865	32.644
	PWM2Vec [22]	NB	0.151	0.926	0.151	0.247	0.234	0.603	109.493		0.528	0.582	0.528	0.441	0.345	0.709	3.033
		MLP	0.529	0.685	0.529	0.403	0.231	0.595	358.965		0.633	0.655	0.633	0.559	0.411	0.695	81.928
		KNN	0.531	0.689	0.531	0.406	0.317	0.625	126.428		0.802	0.796	0.802	0.799	0.623	0.805	0.954
		RF	0.532	0.691	0.532	0.408	0.319	0.625	1052.845		0.842	0.841	0.842	0.836	**0.735**	0.854	10.607
		LR	0.528	0.690	0.528	0.403	0.321	0.626	5643.762		0.679	0.721	0.679	0.670	0.484	0.731	704.796
		DT	0.528	0.691	0.528	0.403	0.321	0.626	142.579		0.819	0.822	0.819	0.816	0.714	0.839	7.636
	Minimizer	NB	0.916	0.989	0.916	0.943	0.801	0.916	90.476		0.490	0.728	0.490	0.430	0.505	0.734	17.223
		MLP	0.952	0.961	0.952	0.948	0.851	0.927	440.944		0.712	0.752	0.712	0.702	0.441	0.709	160.668
		KNN	0.951	0.960	0.951	0.947	0.844	0.926	149.858		0.794	0.798	0.794	0.784	0.576	0.773	3.979
		RF	0.953	0.961	0.953	0.949	0.850	0.927	527.874		0.822	0.831	0.822	0.812	0.710	0.843	54.738
		LR	0.952	0.961	0.952	0.948	0.849	0.927	4918.374		0.799	0.828	0.799	0.786	0.721	0.848	5240.159
		DT	0.952	0.961	0.952	0.948	0.850	0.927	111.393		0.794	0.806	0.794	0.787	0.670	0.826	43.638
Only GANs For Training	Spike2Vec [21]	NB	0.002	0.000	0.002	0.000	0.001	0.491	98.736		0.167	0.035	0.167	0.058	0.016	0.498	12.501
		MLP	0.022	0.032	0.022	0.0192	0.016	0.479	222.003		0.107	0.119	0.107	0.106	0.023	0.511	50.035
		KNN	0.106	0.139	0.106	0.076	0.123	0.558	368.164		0.202	0.082	0.202	0.092	0.035	0.512	4.793
		RF	0.010	0.000	0.010	0.000	0.001	0.500	665.565		0.184	0.033	0.184	0.057	0.016	0.500	26.686
		LR	0.200	0.136	0.200	0.091	0.020	0.500	3497.008		0.053	0.158	0.053	0.048	0.0229	0.487	3809.357
		DT	0.190	0.036	0.190	0.061	0.018	0.499	467.308		0.010	0.000	0.010	0.000	0.001	0.499	36.366
	PWM2Vec [22]	NB	0.026	0.068	0.026	0.003	0.003	0.499	93.458		0.097	0.009	0.097	0.017	0.010	0.500	2.699
		MLP	0.392	0.389	0.392	0.295	0.056	0.499	250.162		0.021	0.190	0.021	0.026	0.012	0.364	39.795
		KNN	0.140	0.205	0.140	0.040	0.016	0.500	343.585		0.092	0.009	0.092	0.016	0.010	0.498	1.421
		RF	0.477	0.227	0.477	0.308	0.038	0.500	644.587		0.318	0.101	0.318	0.153	0.028	0.500	14.304
		LR	0.012	2.070	0.012	4.020	0.001	0.500	4498.689		0.033	0.112	0.033	0.045	0.057	0.490	375.859
		DT	0.002	4.170	0.002	8.324	0.000	0.500	498.689		0.318	0.101	0.318	0.153	0.028	0.500	8.850
	Minimizer	NB	0.023	0.215	0.023	0.035	0.033	0.510	115.915		0.072	0.005	0.072	0.009	0.008	0.500	11.483
		MLP	0.420	0.597	0.420	0.448	0.081	0.514	274.471		0.000	0.000	0.000	0.000	0.000	0.000	63.847
		KNN	0.551	0.690	0.551	0.600	0.152	0.599	382.306		0.176	0.033	0.176	0.056	0.019	0.500	3.665
		RF	0.010	0.000	0.010	0.000	0.001	0.500	792.106		0.179	0.032	0.179	0.054	0.019	0.500	23.71
		LR	0.514	0.235	0.514	0.385	0.047	0.500	3465.703		0.221	0.225	0.221	0.138	0.065	0.521	3603.909
		DT	0.474	0.225	0.474	0.305	0.037	0.500	445.797		0.044	0.015	0.044	0.015	0.006	0.477	39.195

## Data Availability

Not applicable.

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
