# Peer review of "Exploring the Potential of GANs in Biological Sequence Analysis"

_biology, 2023, doi:10.3390/biology12060854_

Round 1
Reviewer 1 Report
The paper proposes using synthetic data generated from clustering to improve clustering for methods that rely on similar-sized clusters, and test a few methods on a few data sets. This seems like a good thing to try, and the results are promising though not clear-cut.
Issues:
I cannot see any attached or linked software and code.
The section "Statistical Significance" is not useful, since it seems to draw five values each from two sets of data and compare them, which breaks the independence assumption of the test and thus merely show that the sets of 5 values are drawn from different datasets, i.e. always true.
While the t-SNE plots and the table are helpful and valuable, something more summarizing is in order, to explain whether the synthetic data generally aids or worsens precision and recall (and accuracy, though I don't understand the word). Preferably with both effect sizes and p-values (calculated as paired tests across data sets, and not counting multiple runs on the same data and method as separate data points).
I find it unclear whether the GAN is fed groups clustered de novo (I assume and hope this is what happened) or use annotated groups.
Reviewer 2 Report
This manuscript proposed to use GANs to synthesize sequence embeddings to balance the training dataset with the goal to improve the sequence classification performance in various traditional machine learning models. First, the authors tried three numerical embedding methods, Spike2Vec, PWM2Vec and Minimizer, to convert biological sequences to vectors ready to be fed into GANs. Four dataset was used in this study to train the GANs with three different approaches: 1. without-GANs synthesized sequences; 2. with-GANs synthesized sequences; 3. only-GANs synthesized sequences. The visualizations of the datasets show that adding GANs synthesized dataset pushes data to form larger clusters in some dataset. A similar phenomenon is also shown in the classification results. I think the authors did a good job of comparing different techniques and generating the results to support the hypothesis that the performance of ML-based classification can be improved by leveraging the strength of GANs. The manuscript is well written in general.
Below are a few comments to help the authors specify confusion and improve the manuscript.
1. lines 67, 74-75, the authors mentioned “significant implications for virus surveillance and tracking” and “development of new antiviral strategies”; however, there is no discussion in the manuscript about how the study could implicate or be applied in these areas. If claimed as contributions, these points need to be thoroughly discussed.
2. line 94, haven’t -> have not
3. lines 102-104, check grammar
4. line 107, don’t -> do not
5. lines 122, 130, 138. justify the choices of k and m. Why not using other values such as k=5 as in other studies? Is the decision based on the characteristics of the specific dataset?
6. line 141, “using the methods mentioned above”. It is not clear at this point how these methods are used. are they used separately or combined?
7. line 141, “to train our GAN model”. missing specifications and justifications on model architecture design. Why such a model is used in this study? More details are required.
8. line 159 “the workflow of GAN is shown in Figure 3”. It is not clear what the workflow is for doing what type of task.
8. Algorithm 1 should be “training GAN model”
9. Figure 3 work flow is not clear. What is it doing?
10. line 168, end sentence with “.”
11. Table 1, last row, why sequence lengths vary in such a big range? what are the effects?
12. line 178, cite palmscan
13. line 185, define TRBs
14. line 189, provide the github link with a clear manual.
15. line 199. “the addition of GANs-based features”. At this point, it is not clear what you mean by “addition”. I’d suggest moving the paragraph from lines 255 to 261 to an earlier section when the “with-GAN” approach was first introduced.
16. Figure 4. use contrast colors. Why H3N2 in (b) has way fewer data points than H1N1 comparing to (a) and (c)?
17. section 4.2. In addition to visualizing the dataset, the authors could consider to provide numeric cluster evaluations, i.e. various cluster indices, to make direct comparisons of clusters.
18. line 260. it’s not clear if the test set is balanced or imbalanced as an imbalance test set also has an impact on accuracy.
19. Tables 5 and 6, highlight the “best” performers in each category.
20. Section 5.4, list results
Please check grammars. i.e. lines 94, 107, 102-104, 168, 262, etc.
